# Environmental controls on ecosystem-scale cold season methane and carbon dioxide fluxes in an Arctic tundra ecosystem.

Dean Howard[a,b], Yannick Agnan[c], Detlev Helmig[a], Yu Yang[d], and Daniel Obrist[b]

[a]Institute of Arctic and Alpine Research, University of Colorado Boulder, Boulder, CO, USA
[b]Department of Environmental, Earth and Atmospheric Sciences, University of Massachusetts Lowell, Lowell, MA, USA
[c]Earth and Life Institute, Université Catholique de Louvain, Louvain-la-Neuve, Belgium
[d]Department of Civil and Environmental Engineering, University of Nevada, Reno, NV, USA

**Correspondence:** Daniel Obrist (Daniel_Obrist@uml.edu)

**Abstract.** Understanding the processes that influence and control carbon cycling in Arctic tundra ecosystems is essential for making accurate predictions about what role these ecosystems will play in potential future climate change scenarios. Particularly, air–surface fluxes of methane and carbon dioxide are of interest as recent observations suggest that the vast stores of soil carbon found in the Arctic tundra are becoming more available to release to the atmosphere in the form of these greenhouse gases. Further, harsh wintertime conditions and complex logistics have limited the number of year-round and cold season studies and hence too our understanding of carbon cycle processes during these periods. We present here a two-year micrometeorological data set of methane and carbon dioxide fluxes, along with supporting soil pore gas profiles, that provide near-continuous data throughout the active summer and cold winter seasons. Net emission of methane and carbon dioxide in one of the study years totalled 3.7 and 89 g C m$^{-2}$ a$^{-1}$ respectively, with cold season methane emission representing 54% of the annual total. In the other year, net emission totals of methane and carbon dioxide were 4.9 and 485 g C m$^{-2}$ a$^{-1}$ respectively, with cold season methane emission here representing 82% of the annual total – a larger proportion than has been previously reported in the Arctic tundra. Regression tree analysis suggests that, due to relatively warmer air temperatures and deeper snow depths, deeper soil horizons – where most microbial methanogenic activity takes place – remained warm enough to maintain efficient methane production whilst surface soil temperatures were simultaneously cold enough to limit microbial methanotrophic activity. These results provide valuable insight into how a changing Arctic climate may impact methane emission, and highlight a need to focus on soil temperatures throughout the entire active soil profile, rather than rely on air temperature as a proxy for modelling temperature–methane flux dynamics.

## 1 Introduction

Active-layer soils and permafrost soils in the Arctic permafrost region contain significant stores of terrestrial organic carbon. These ecosystems account for an estimated 1307 (1140–1476) Pg of organic carbon, with ∼1035 Pg found within soils between 0 and 3 m depth (Hugelius et al., 2014). Recently-observed increases in surface air temperature within these regions (Polyakov et al., 2002) have sparked interest in the biogeochemical cycling of this carbon store, as substrate metabolic activity — shown to be positively correlated to temperature — can break down organic compounds in the soil, releasing soil organic carbon to

the atmosphere in the form of carbon dioxide and methane (Lai, 2009). Furthermore, the "active layer" horizon, within which most soil carbon decomposition takes place, has been observed in places to be expanding as the underlying permafrost thaws under the influence of a warming atmosphere, thus exposing larger quantities of organic carbon to decomposition (Schuur et al., 2009; Romanovsky et al., 2010; Schuur et al., 2015; Vonk and Gustafsson, 2013).

Production of methane in the carbon-rich soils of the Arctic tundra takes place as a result of microbial metabolic activity (Lai, 2009). The break-down of organic carbon to form methane is a complex process that requires contributions from various taxa of microorganisms (Whalen, 2005). Methanogens form the last step in this process of producing methane from organic polymers (Conrad, 1999); whilst these methanogens encounter oxygen in the breaking down of acetate and carbon dioxide, they cannot survive in oxic environments (Whalen, 2005; Kamal and Varma, 2008; Lai, 2009). As such, methanogenesis in peatlands is an obligate anaerobic process that takes place largely within deeper, anoxic layers of the soil, generally below the water table level (Le Mer and Roger, 2001). Methane production at these depths creates concentration gradients that lead to upward diffusion of methane through the soil to the surface (Preuss et al., 2013). Furthermore, methane can be transported to the surface via ebullition and through aerenchymatous tissues within some vascular plants (Joabsson et al., 1999; Lai, 2009).

In the upper soil layers, another subset of microorganisms known as methanotrophs consume a portion of this produced methane in the presence of oxygen, eventually oxidising it to carbon dioxide (Lai, 2009). These methanotrophs are largely exposed to methane diffusing through the soil pore space, as ebullition is too quick to allow exposure to methanotrophs (Boone, 2000) and plant vascular transport shields methane from methanotrophic activity (Schimel, 1995; King et al., 1998; Verville et al., 1998). The rate of methane consumption is usually highest immediately above the water table level, where high concentrations of methane formed from the underlying methanogens meet with sufficient oxygen levels from the overlying atmosphere (Dedysh et al., 2002). Rates of both methanogenesis and methanotrophy are highly dependent on temperature, with optimal metabolic rates (as determined in the laboratory) occurring at temperatures of around 25 °C (Dunfield et al., 1993). Of the two competing processes, methanogenesis has shown to be more temperature-dependent with higher reported rate changes per unit warming (e.g. per 10 °C ($Q_{10}$): 5.3–16) compared to methanotrophy ($Q_{10}$: 1.4–2.1) (Dunfield et al., 1993).

Estimating the methane exchange budget in Arctic tundra ecosystems and how it relates to temperature are challenging objectives, currently subject to considerable uncertainties. Ambient observations in northern Alaska over 29 years showed no clear increase in ambient atmospheric methane concentration enhancements during this period, despite noticeably warmer air temperatures (Sweeney et al., 2016). Direct observations of methane exchange, however, during the Carbon in Permafrost Experimental Heating Research (CiPEHR) project, showed significant increases in methane emission under warming soil conditions (Natali et al., 2015). Multi-year carbon exchange data sets are rare, and challenging winter conditions at Arctic sites has led to many studies focussing largely on summer and early autumn periods (Euskirchen et al., 2012). The first year-round micrometeorological methane exchange measurements reported showed that cold season methane emission dominated the annual exchange budget, suggesting a predominant role of cold-season processes (Zona et al., 2016). At five Alaskan sites (four on coastal plains and tundra dominated by sedges, grasses, and mosses within the northern coastal region surrounding Utqiaġvik and one over tussock-sedge dwarf-shrub, moss tundra at Ivotuk on the North Slope of the Brooks Range, Walker et al., 2005), substantial methane emission was reported to have occurred throughout the cold season, and although emission

rates were lower than those measured during the warmer summer period, prolonged wintertime activity amounted to 50% ± 9% (mean ± 95% confidence interval) of annual emission. The authors suggest that this cold season emission may become more important under forecasted climate conditions that include higher air temperatures and deeper snowpacks (Hay and McCabe, 2010; Zona et al., 2016).

We present here a two-year micrometeorological methane and carbon dioxide exchange data set, undertaken over an acidic tussock tundra site near the Toolik Field Station, Alaska, USA, on the north slope of the Brooks Range. Complimentary to air–surface exchange measurements, we report soil pore space methane, carbon dioxide and oxygen concentrations and soil water content in the upper 40 cm, as well as soil temperature profiles at and near the site to a depth of 150 cm. All measurement systems were deployed year-round, providing near-continuous data coverage throughout both the summer growing seasons and the cold winter seasons in both years. The goal of this study was to investigate environmental controls that significantly impact the magnitude and direction of methane fluxes in this environment – particularly over the colder winter months – as environmental control–methane flux relationships during these periods are relatively poorly understood. These results expand our coverage of year-round methane and carbon dioxide exchange data sets across different bioclimates and landscapes, as well as add to our growing understanding of carbon exchange dynamics in the Arctic tundra soils, and provide insights into how these dynamics may evolve under forecasted changing conditions in this region.

## 2   Methods

### 2.1   Site description

This study was performed over two full years from October 2014 to September 2016 at Toolik Field Station (Alaska, USA) located on the north slope of the Brooks Range (68° 38' N, 149° 36' W, 720 m a.s.l.). The study site, approximately 180 km south of the Arctic Ocean, overlies Cretaceous shale, claystone, siltstone, and sandstone, with soils that are characterised as cryosols. Vegetation within the measurement footprint (see Fig. S1) is dominated by an acidic tussock tundra vegetation: scrubby plants (e.g. *Cassiope tetragona* (L.) D.Don, *Arctostaphylos alpinus* (L.) Spreng.), shrubs (e.g. *Betula nana* L., *Salix pulchra* Cham.), tussock grasses (*Carex*), mosses and lichens (Shaver and Chapin III, 1991). Vegetation in other areas of the measurement footprint were characterised as wet graminoid tundra (sedge and moss tundra). Both mineral and organic soil profiles were closely present together with different horizon depths. Soil organic carbon content, based on eight soil sampling pits to a depth of 90 cm around the flux tower, was highly variable, with A-horizon organic carbon concentrations averaging 10.3% (range of 7 to 14%) and B-horizon organic carbon concentrations averaging 2.4% (range of 1 to 4%) (Olson et al., 2018). Estimates of soil organic carbon stores in the flux footprint range from 5 to 25 kg C m$^{-2}$ (Fig. S2).

### 2.2   Instrumentation

An aerodynamic gradient approach was utilised for observing air–surface methane and carbon dioxide fluxes. The aerodynamic gradient method was chosen for compatibility with a concurrent atmospheric mercury flux study (Obrist et al., 2017). This

method has been shown to have greater variability compared to the more widely-used eddy covariance method over diel time scales (Muller et al., 2009) though with reasonable agreement over longer time scales, with the caveat that the concentration gradients are precisely quantified using high-precision gas analysers (Zhao et al., 2019; Karlsson, 2017; Fritsche et al., 2008). Turbulent characteristics were measured using a Metek USA-1 sonic anemometer (Metek GmbH, Elmshorn, Germany), positioned 2.36 m above the tundra soil. Atmospheric sampling of trace gas concentrations was performed at heights $z_1 = 0.61$ m and $z_2 = 3.63$ m. In addition to atmospheric trace gas sampling, a soil trace gas system consisting of four to six soil inlets in two vertical profiles installed between 15 and 20 m to the north of the flux tower (herein "flux tower profiles") allowed for determination of near-surface trace gas concentration gradients in soil pore air (Fig. S3). These profile locations were chosen to represent an organic matter-rich soil profile (average carbon content 10.6% in the A-horizon and 6.9% in the B-horizon, herein "organic") and a less organic matter-rich profile (13.1% and 2.5% carbon contents in the A- and B-horizons, herein "mineral"). Data from these two soil inlet profiles were collected at two depths in Year 1 (10 and 40 cm), and three depths in Year 2 (10, 20 and 40 cm). Perfluoroalkoxy Teflon tubing from both the atmosphere and soil inlets were directed in a heated conduit to an onsite field laboratory, and a solenoid valve system allowed sequential sampling between all inlets with switching interval of 10 minutes (Obrist et al., 2017). Methane and carbon dioxide concentrations were quantified using a Los Gatos 915-0011 ultra-portable greenhouse gas analyser (Los Gatos Research, Mountain View, CA, USA), factory-calibrated prior to installation and operating at 1 Hz. The analyser was zeroed using methane- and carbon dioxide-free zero air approximately every 6 weeks. Span calibrations were achieved using the internal calibration routine, as recommended by the manufacturer. Line intercomparison tests were also performed with the same frequency by moving both inlets to the same height and sampling for between 12 and 24 hours (average 17 hours). Concentration differences between the sample lines during the intercomparisons (null gradients) were <0.001 $\mu$mol mol$^{-1}$ (or 0.025% of the mean concentration gradient) for methane (mean methane gradients were -4 $\pm$ 7 $\mu$mol mol$^{-1}$) and <0.1 $\mu$mol mol$^{-1}$ (0.1% of the mean concentration gradient) for carbon dioxide (mean carbon dioxide gradients were 100 $\pm$ 200 $\mu$mol mol$^{-1}$). One-way ANOVA tests performed on the line intercomparison data showed that methane null gradients were not significantly different throughout both years ($p = 0.03$), however this was not the case for carbon dioxide null gradients, with those from the final intercomparison (in December 2015) being significantly lower than the rest. Estimation of the cumulative uncertainty calculated from null gradient data (achieved by substituting $(C_2 - C_1)$ in Eq. 1 with $(C_2 - C_1 + \epsilon)$, where $\epsilon$ is the measured null gradient value), gave values of 0.5% and 0.2% for methane in Years 1 and 2, respectively. For carbon dioxide, the respective uncertainties were 8% and 47%. Although the emphasis here is on methane flux magnitudes and dynamics, carbon dioxide fluxes are discussed at length in order to understand corresponding respiration processes that help us constrain the influence of microbial activity on observed methane fluxes.

Within the flux tower soil profiles, temperature and volumetric water content (VWC) were also measured at depths of 10, 20 and 40 cm. Temperatures were measured using soil temperature probes (Model 107, Campbell Scientific Inc., Logan, UT, USA) and VWC was measured at the same depths using time-domain reflectometry (Model CS615-L Soil Volumetric Water Reflectometers, Campbell Scientific Inc., Logan, UT, USA). Further from the flux tower (430 m to the north-east), Toolik Field Station operates two profiles of soil temperature (thermocouple) measurements to a depth of 150 cm (0, 5, 10, 20, 50, 100 and 150 cm). Despite the increased distance, these profiles were included in the current analysis as they provide a longer time

series (measurements have been taken continuously since 1988) and information at deeper depths than the flux tower profiles. A snow tower (Seok et al., 2009; Faïn et al., 2013) was installed prior to the first snowfall and recorded temperatures at 0, 10, 20, 30, 40 and 110 cm above the soil surface, thus measuring temperatures within the snowpack as it developed above each measurement height. The average snowpack depth over the site was measured daily using a camera set to automatically record images of reference snow stakes (Agnan et al., 2018). These depth measurements began in November 2014, and so the first snowfalls in that year were not recorded.

## 2.3 Calculations

Fluxes of methane and carbon dioxide were calculated using the aerodynamic gradient approach described by Edwards et al. (2005):

$$F = \frac{-ku^*(C_2 - C_1)}{\ln((z_2 - d)/(z_1 - d)) - \Psi_2 + \Psi_1},$$

(1)

where $F$ represents the flux of either methane or carbon dioxide, $k$ the von Kármán constant, $u^*$ the friction velocity, $C_i$ the concentration of atmospheric trace gas species in question at height $i = [1, 2]$, $z_i$ the sampling height, $d$ the displacement height and $\Psi_i$ the stability-dependent integrated similarity functions for heat, as given by Businger et al. (1971), that are well-represented for scalars within a range of Obukhov lengths between -2.5 and 2. Values outside of this range (highly stable or highly unstable) were filtered from analysis, resulting in a loss of 1.1% of available data. Additionally, periods for which the friction velocity was below 0.08 m s$^{-1}$ (Muller et al., 2009) were excluded from analysis. This resulted in a loss of 7.1% of all available data, with a slight seasonal bias (7.1% of winter (DJF) data compared to 2.9% of summer (JJA) data). Herein we follow the convention of positive flux values representing emission, whilst negative values represent deposition.

Atmospheric turbulent characteristics (friction velocity and Obukhov stability) were calculated using the flux processing software EddyPro v.6.2.0 (Li-COR, Lincoln, NE, USA) using 30-minute averaging periods. Rotation of sonic data into mean wind vectors was accomplished using the double rotation technique and quality control tests for steady state and developed turbulent conditions were implemented according to Foken et al. (2004). Apparent sonic anemometer sampling height was altered according to daily observed snow depth in increments of 5 cm, as were gradient intake sampling heights. Displacement height $d$ was set as $0.7h_c$ (canopy height, ∼0.2 m) during snow-free periods and at 0 m during snow-covered periods (Oke, 1978). As a single instrument was used for trace gas sampling at both intake heights, leading to a loss of temporal coverage within each 30-minute period (Woodruff, 1986), gaps in the concentration time series were estimated for each averaging period using a 4th order polynomial fit to the observed concentration time series. Average concentrations at each height were then calculated from a truncated mean (10th–90th percentile) in order to reduce effects of outliers (Fig. S4).

Two-dimensional footprint analyses were undertaken for each 30-minute period using the method of Kljun et al. (2015) and fluxes for which the footprint intensity over the adjacent Toolik Lake was shown to be >20% of the total were removed from analysis. Analysis of energy balance closure showed that calculated turbulent and soil heat fluxes for snow-free periods, excluding fetches in the direction of Toolik Lake, accounted for approximately 88% of net radiative fluxes (linear least

squares, $p < 0.001$). Gaps in both methane and carbon dioxide fluxes, resulting from quality control and instrument downtime/maintenance, were filled using the MDSGapFill function (Reichstein et al., 2005) within the R package REddyProc (Wutzler et al., 2018). The efficacy of this gap filling was tested against a randomly-selected validation set of size equal to 10%

of available flux values (Fig. S5). Ecosystem respiration was approximated using half-hourly, gap-filled carbon dioxide fluxes, filtered to exclude times when incoming photosynthetically active radiation (PAR) was above 5 $\mu$mol m$^{-2}$ s$^{-1}$ (Natali et al., 2015). Periods during which no data fitting this criteria are available (i.e. polar day) were not gap-filled, resulting in incomplete temporal coverage for ecosystem respiration. Regression tree analysis (Sachs et al., 2008; De'ath and Fabricius, 2000; Breiman et al., 1984) was undertaken on 80% of observed flux data (20% validation fraction) using the TreeBagger function in Matlab

2016a (MathWorks, Natick, MA, USA). 500 cross validations were ran with a minimum leaf size of 1% of the training set size, with the tree with lowest mean squared error chosen as our predictive model.

## 3 Results and Discussion

### 3.1 Site climatology

Air temperatures, as observed at Toolik Field Station, showed similar patterns and magnitudes between 2014–15 and 2015–16

(Fig. S6), ranging between -40 °C and 0 °C during the winter months (DJF) and -5 °C and 20 °C during the warmest summer months (JJA). Air temperatures in these years also remained within the range of those observed during the preceding 26 years during all months of the year. Students $t$-tests could not reject the null hypothesis that mean air temperature values for winter and summer months in the study years came from a different distribution than the climatological record ($p = 0.26$ and 0.68, respectively). Soil temperatures at 20 cm depth were within the expected climatological range during the summer months,

however throughout the colder months (between mid-November and late April), temperatures during 2014–15 and 2015–16 were among the warmest observed since 1988 (Student's $t$-test, $p < 0.001$). Likewise, at 100 cm depth, wintertime soil temperatures during 2014–15 and 2015–16 were among the warmest seen in the Toolik Field Station record (Student's $t$-test, $p < 0.001$). Minimum cold season soil temperatures at these depths for 2014–15 and 2015–16 were the second- and third-highest on record, respectively (Fig. S7). To investigate the climatological influence of atmospheric forcing on soil cooling,

Fig. S7 also shows that 2014–15 had the shortest cold season (defined in this instance as the period during which the 28-day running mean of 5 m air temperature remains below 0 °C) and the third-smallest freezing degree day (FDD) value on record. Whilst 2015–16 had an average-length cold season (20th shortest on record), it had the fourth-smallest FDD value since 1988.

Though snow depth has not been measured at Toolik Field Station across the same period of time, observations at the snow tower provide some additional insights into why soils were warmer during the winter of 2014–15 compared to 2015–16 (Fig.

S8). Snow depth was significantly (Student's $t$-test, $p < 0.001$) larger in 2014–15 (mean 32 cm) than in 2015–16 (mean 22 cm). Deeper snowpacks are able to provide an increased thermoinsulation effect from cold air temperatures, particularly in the early cold season (Maksimova et al., 1977; Sokratov and Barry, 2002), thus leading to warmer and less variable surface soil temperatures. This effect can be seen in the temperature pulses shown in the snow tower thermocouple data (Fig. S8) and their

effect on the underlying soil, where the minimum subnivean surface temperature in 2014–15 (-12 °C) was 5 °C warmer than that observed in 2015–16 (-17 °C).

Arctic tundra ecosystems are highly heterogeneous within the scale of micrometeorological flux footprints (typically 10s to 100s metres, Kljun et al., 2015; Fox et al., 2008), and during the winter, the combined effects of wind and topography lead to even greater spatial heterogeneity in snow depths and snow physical properties (Agnan et al., 2018). Sub-surface soil temperatures, which are further influenced by air temperature and downwelling radiation; overlying vegetation and snow; and soil properties and moisture, are likely highly spatially variable within the footprint as well. As a result, the limited number of soil temperature profiles within the flux measurement footprint may not be fully representative of the average temperature within the flux footprint. The four soil temperature profiles we have available are separated both in space and in the soil properties in which they were installed. Fig. S9 shows the time series for each of these profile measurements at two common depths (10 and 20 cm). This time series shows that, whilst the absolute range of temperatures between measurements can be pronounced, the correlation between these soil temperature measurements is reasonable enough throughout most of the study period (mean $R^2$ = 0.64) to warrant use of Toolik Field Station data for further investigation of temporal trends. The decision to use Toolik Field Station data was driven primarily by the deeper profiles measured here, as this information is vital to the primary outcomes of this study (see Section 3.4).

## 3.2   Annual and seasonal flux patterns

Half-hourly methane and carbon dioxide fluxes are shown in Figure 1, along with total cumulative values over the ∼two-year (727-day) study. Overall, based on combined raw and gap-filled data, the mean half-hourly methane flux showed an emission of $0.5 \pm 0.5$ mg C m$^{-2}$ h$^{-1}$ (herein, uncertainty is expressed as one standard deviation of the measured values). The distribution of methane fluxes (Fig. S10) showed a positive skew, as well a secondary peak in values close to zero. Cumulative diel sums gave a mean net daily flux of $11 \pm 8$ mg C m$^{-2}$ d$^{-1}$. Over the study period as a whole the site acted as a net source of methane, with a cumulative methane emission of 8.6 g C m$^{-2}$ (4.9 g C m$^{-2}$ and 3.7 g C m$^{-2}$ in the first and second years, respectively). The overall mean carbon dioxide flux across the two years of measurements was $0.0 \pm 0.2$ g C m$^{-2}$ h$^{-1}$ (mean net daily flux of $1 \pm 3$ g C m$^{-2}$ d$^{-1}$), with the site acting as a net source of carbon dioxide during both years of measurements. The distribution of carbon dioxide fluxes across the study (Fig. S10) did not show skewness as seen in the methane flux distribution, though it did show a higher level of kurtosis. During the 24-month measurement period, the site emitted a net carbon dioxide flux equivalent to 583 g C m$^{-2}$ (485 g C m$^{-2}$ and 89 g C m$^{-2}$ in the first and second years).

Seasonality can be defined in a number of different ways depending on the processes of interest (Mastepanov et al., 2013); initially, we followed similar definitions as those described by Zona et al. (2016) who investigated changes in methane fluxes based on surface (10 cm) soil temperatures. Periods where surface soil temperatures were above 0 °C were defined as "active" seasons (yellow shading in Fig. 1) and those where soil temperatures were below 0 °C were defined as "frozen" seasons (blue shading in Fig. 1). Zero curtain periods, where surface soil temperature remains close to 0 °C ($\pm$ 0.5 °C) for prolonged time periods due to latent heat released or absorbed from soil water, were separated into "freezing" or "thawing" seasons. Freezing seasons (turquoise shading, Fig. 1) occurred prior to the frozen season, whilst thawing seasons (green shading, Fig.

1) occurred after the frozen season and prior to the active season. Combined freezing–frozen–thawing periods were defined as "cold season". The period from the onset of the freezing season in 2014 until the end of the active season in 2015 has herein

been defined as "Year 1", whilst the same seasons from 2015 until 2016 have been defined as "Year 2". Tables 1 and 2 give summary methane and carbon dioxide flux data, respectively, for these seasons and years so defined.

Table 1 shows marked differences in the magnitude and seasonality of methane fluxes between the two years. Cumulative methane emission in Year 1 was 1.3-fold that of Year 2, across a slightly shorter period (347 days compared to 380 days). All seasons showed net methane emission across the study period, with statistically significant differences (Student's two sample

$t$-test, $p <0.05$) in the net daily flux between Year 1 and Year 2 for all seasons. The largest seasonal differences in net daily methane fluxes between years were for the active and frozen seasons (6 and 10 mg C m$^{-2}$ d$^{-1}$, respectively). For Year 2, the active season showed significantly higher methane emission compared to Year 1, releasing 1.7 g C m$^{-2}$ (15 mg C m$^{-2}$ d$^{-1}$), or 46% of the annual total, compared to 0.9 g C m$^{-2}$ (9 mg C m$^{-2}$ d$^{-1}$), or 18% of the annual total for Year 1. Conversely, the frozen season showed higher emission in Year 1, releasing 2.5 g C m$^{-2}$ (16 mg C m$^{-2}$ d$^{-1}$), or 51% of the annual total,

compared to 1.1 g C m$^{-2}$ (6 mg C m$^{-2}$ d$^{-1}$), or 30% of the annual total in Year 2. Year 1 also showed higher methane emission in the freezing and thawing seasons, though these represented similar percentages of the annual total across both years ($\sim$25% of annual total for freezing and 3–4% of annual total for thawing). Zona et al. (2016) reported average cold season methane emission from five Alaskan Arctic sites of $1.7 \pm 0.2$ g C m$^{-2}$, accounting for between 37 and 64% of the total annual methane budget at these sites. The authors note that these contributions are higher than those estimated from previous models and

periodic chamber observations. In our study, observations in Year 2, where 50% of annual methane emission occurred in the cold season, are within the ranges reported by Zona et al.. However, cold-season methane emission during Year 1 accounted for 82% of annual net emission, indicating that cold-season methane emission can strongly dominate annual flux magnitudes, to a larger extent than recent evidence suggests.

Figure 1a shows the detailed temporal patterns that help explain differences in seasonal net emission between the two years.

For Year 1, the 28-day moving average (herein MA28) methane flux (red line in Fig. 1a) was initially relatively high and positive at the onset of the freezing season ($\sim$1 mg C m$^{-2}$ h$^{-1}$), and remained at a similarly high level during the Year 1 freezing season and most of the frozen season. In early March 2015, the MA28 methane flux began to steadily decline during the late frozen season, thawing season, and mid-way into the active season, reaching a minimum emission of $\sim$0.1 mg C m$^{-2}$ h$^{-1}$ in August 2015 before increasing again to $\sim$0.7 mg C m$^{-2}$ h$^{-1}$ at the onset of the Year 2 freezing season in September 2015. Soil methane

concentrations in the organic profile (Fig. S3) support evidence of methane emission throughout the Year 1 cold season (until the inlets were flooded in the Year 1 thawing season), with elevated concentrations relative to the atmosphere and an increasing concentration gradient with depth. This evidence is however not visible in the mineral profile data, with methane concentrations within the upper 40 cm remaining close to atmospheric concentrations. In contrast to Year 1, in the Year 2 freezing season, MA28 methane flux began to decline in October 2015, and continued a consistent and relatively constant decline throughout

the winter until it approached $\sim$0.2 mg C m$^{-2}$ h$^{-1}$ in February 2016. It was not until the thawing season in June 2016 that the MA28 methane flux again began to increase in magnitude (net positive), reaching a peak of $\sim$0.9 g C m$^{-2}$ h$^{-1}$ about mid-way through the Year 2 active season. Again, methane concentrations in the organic profile support evidence of continued methane

emission throughout the cold season, though the concentration gradient is not as steep as in Year 1. Large concentration spikes in the early freezing season and thawing season are likely due to decreases in soil permeability that "trap" methane from sources below. Throughout this period the mineral soil profile again shows little evidence of methane emission, and rather shows evidence of a methane sink in the freezing and thawing seasons.

Carbon dioxide fluxes (Table 2) showed significant net emission throughout the entire cold season in Year 1 (471 g C m$^{-2}$ or 1.9 g C m$^{-2}$ d$^{-1}$), followed by minor net emission in the subsequent active season (14 g C m$^{-2}$ or 0.1 g C m$^{-2}$ d$^{-1}$). Year 2 freezing and frozen seasons showed lower carbon dioxide emission than Year 1, and the thawing season showed net carbon dioxide deposition, resulting in a combined cold season emission of 294 g C m$^{-2}$ or 1.1 g C m$^{-2}$ d$^{-1}$ (38% lower than Year 1). Net active season carbon dioxide fluxes in Year 2 showed significant deposition, at -211 g C m$^{-2}$ or -1.9 g C m$^{-2}$ d$^{-1}$. Annual and multi-season micrometeorological flux studies are rare for the Alaskan Arctic (Commane et al., 2017); however, the net annual carbon dioxide flux for Year 2 is within the range of values reported for wet sedge tundra (2 to 147 g C m$^{-2}$ a$^{-1}$, Euskirchen et al., 2012, 2017), and larger than for heath tundra (21 to 61 g C m$^{-2}$ a$^{-1}$, Euskirchen et al., 2012, 2017) or tussock tundra (13 to 15 g C m$^{-2}$ a$^{-1}$, Euskirchen et al., 2012; Oechel et al., 2014). The low end of the range of values quoted for wet sedge tundra (2 g C m$^{-2}$ h$^{-1}$, Euskirchen et al., 2012) is based on a period when active season deposition largely balanced cold season emission; Euskirchen et al. (2017) report in their longer-term study of this wet sedge site a trend towards larger annual net emission values that are largely attributed to increasing cold season emission, with little trend seen for active season deposition. They note an increasing trend in September–December carbon dioxide emission of 1.34 g C m$^{-2}$ for each additional day of zero curtain (freezing season) length. Here, the observed difference was much larger, with an additional 126 g C m$^{-2}$ loss observed in Year 1 over a 10 day longer freezing season (Table 2). The net annual carbon dioxide flux for Year 1 was significantly above the range previously reported for wet sedge tundra. Arctic tundra ecosystems are highly heterogeneous both physically and biogeochemically (Fox et al., 2008) and the area examined here is no exception. Seasonal two-dimensional footprint analyses (Fig. S11) showed a prdeominantly southerly footprint during all seasons, where fens and moist tundra are more abundant (Fig. S1). Importantly, the homogeneity in all seasonal footprints shown in Fig. S11 excludes the possibility that observed differences in seasonal flux magnitudes (i.e. higher cold season contributions in Year 1 relative to Year 2) are due to flux footprint differences over heterogeneous surfaces.

Figure 1b shows the detailed temporal patterns of seasonal net carbon dioxide emission for the two years. Active season net carbon dioxide sinks in both years are consistent with long-term eddy covariance observations that show summertime sink trends in the Alaskan Arctic (Oechel et al., 2008). These active season carbon dioxide sinks also are consistent with large-scale aircraft observations over the Alaskan North Shore tundra (Commane et al., 2017) that show a switch to carbon dioxide deposition during June–August, peaking at approximately -0.3 g C m$^{-2}$ h$^{-1}$. For Year 1 however, the active season sink was relatively short-lived, resulting in a net active season emission (0.9 g C m$^{-2}$ h$^{-1}$) that contributed to the large net annual emission for Year 1. Cold season MA28 carbon dioxide fluxes are almost consistently positive, only switching to net uptake in the late thawing season of Year 1. A major distinction between the two years of measurements is the duration of relatively high emission – also noted by Commane et al. – that begins in the late active season and extends into the freezing season. As shown in Figure 1b, while MA28 carbon dioxide emission decreased relatively early in Year 2, dropping below 0.1 g C m$^{-2}$ h$^{-1}$

in October 2016, in Year 1 this decline occurred at a slower rate, remaining relatively high and not reaching the same low value until January 2015. Soil concentrations in the organic profile similarly show a sustained carbon dioxide concentration gradient in the upper soil that persists throughout the Year 1 freezing season, whilst for Year 2 the concentration gradient shrinks to near-zero in January. This contraction of the carbon dioxide concentration gradient in Year 2 is mirrored in the mineral soil profile, whilst in Year 1 the mineral profile shows almost no concentration gradient in the Year 1 cold season. In a latitudinal comparison from three Alaskan tundra sites, Grogan and Chapin III (1999) reported significantly higher wintertime carbon dioxide efflux from Toolik, relative to Fairbanks (south of Toolik) and Sagwon (north of Toolik). This wintertime carbon dioxide efflux was correlated with warmer surface soil temperatures (quantified as 5 cm soil temperatures greater than -5 °C), which at Toolik were relatively high due to thermal insulation by a substantial early snowfall. The extended period of increased carbon dioxide emission in the Year 1 freezing season is likely also associated with the insulating effects of an early substantial snowfall and the associated warmer surface soil temperatures (explored in greater detail in the following section). Importantly, in both years, MA28 carbon dioxide fluxes do not completely cease during the cold season and always maintain a small emission throughout winter (up to 0.1 g C m$^{-2}$ h$^{-1}$).

In summary, the two years showed substantial temporal differences in methane fluxes, with Year 1 showing higher methane emission throughout most of the cold season (100% greater), contributing a high fraction (82%) of annual net methane emission. This is in contrast to Year 2, which experienced a continued decline in methane emission that began early in the freezing season, resulting in a relatively low contribution (54%) to the annual total. Annual carbon dioxide flux magnitudes were most similar to other wet sedge tundra measurements; show strong seasonal trends with relatively high respiration in the freezing season and prolonged but low carbon dioxide emission in the frozen season; and carbon dioxide deposition during the thawing and/or active season, largely in agreement with carbon dioxide flux patterns reported for northern tundra ecosystems before. Inter-annual comparison, however, showed cold-season carbon dioxide fluxes that were 38% higher in Year 1, also largely driven by slower and later declines in carbon dioxide emission fluxes during the freezing and frozen periods.

## 3.3 Soil temperature relationships

Continuous cold season methane flux data at the ecosystem level are rare for Arctic tundra ecosystems. Given the strong dominance of cold-season fluxes for annual flux magnitudes (54% to 82% in our study), the pronounced differences in cold season methane flux dynamics between the two years merit particular attention. Methane flux dynamics are controlled largely by soil methanogenic and methanotrophic activity (Lai, 2009), and previous research has suggested that, in frozen soils (where water table dynamics become less important), soil temperature has the strongest control on microbial activities that drive methane production and consumption (Le Mer and Roger, 2001). In order to maintain comparability with similar year-round methane flux observations in the Alaskan Arctic (Zona et al., 2016), our initial investigations into relationships between ecosystem-level methane fluxes and underlying soil temperatures began with soil temperatures measured at 10 cm depth (hereafter surface soil temperature). This is shown in the upper panels of Figure 2, where MA28 methane fluxes are plotted against MA28 surface soil temperatures, as measured at the Toolik Field Station. Horizontal lines show the spread of all available surface soil measurements (i.e. both at Toolik Field Station and closer to the flux tower).

Based on these upper panels in Figure 2, it is evident that methane fluxes showed very different relationships with surface soil temperature in Year 1 compared to Year 2. For Year 2 (Fig. 2b), we observed a pattern similar to that reported by Zona et al. (2016), whereby MA28 methane fluxes began to decrease mid-way through the freezing season (point F, October 2015). As surface soil temperatures continued to decrease during the frozen season (F–G), MA28 methane fluxes continued to decline, reaching a minimum during the frozen season in March 2016 (point G). MA28 methane fluxes remained low (between 0.1 and 0.2 mg C m$^{-2}$ h$^{-1}$) during the remaining frozen season, as MA28 surface soil temperatures increased from its minimum of -5.1 °C to 0 °C (G–H). As surface soil temperatures continued to warm above 0 °C during and beyond the late thawing season, methane emission increased significantly into the active season (H–I), peaking in August 2016 (point J). The relationships observed in Year 2 provide evidence that surface soil temperature is correlated with cold season methane fluxes, yet with substantial variability and a strong hysteresis between freezing and thawing periods. Zona et al. (2016) suggested that temperature-dependent decreases in the near-surface methane oxidative capacity were largely responsible for the slow attenuation of methane fluxes in the early frozen period (here F–G), noting that sites with the largest and warmest active layers displayed the slowest decrease in methane fluxes. Soil pore gas concentration measured in our study show that oxygen levels within the upper 40 cm were sufficient (>17%) to ensure methane oxidation in this zone across the entire study period and hence that the top 40 cm soils were continuously oxic (Fig. S3). We also note that soil pore gas measurements took place in an elevated, drier tussock region and that the thickness of the upper oxidative region is expected to be smaller in lower depression areas and more water-saturated wet sedge regions (Gebauer et al., 1996). Due to the dominance of methanotrophic microbial communities in the upper oxic region, surface soil temperatures are likely an underlying reason, and hence a good predictor, for observed declines in freezing season methane emission. However, the strong Year 2 active season increase in methane emission is unlikely causally related to surface soil temperatures, as methanogenic microbial communities are less abundant in the upper 40 cm of soils. Instead, as suggested by Zona et al., increases in methane emission as surface soils warm above 0 °C (points H–I in Fig. 2b) are likely coincident with enhanced methanogenic activity as temperature pulses reach deeper anoxic soil layers (see Section 3.4).

In contrast to Year 2, Year 1 MA28 methane fluxes showed almost completely reversed relationships with surface soil temperatures (Fig. 2a). MA28 methane fluxes remained high (mean 0.8 mg C m$^{-2}$ h$^{-1}$) throughout the freezing and early frozen seasons in spite of temperatures decreasing below freezing (A–B), with values among the largest observed across the entire study period. Methane fluxes only began to decrease after March 2015 (point B) as MA28 surface temperature approached its minimum value of that season (-2.6 °C). Thereafter, methane fluxes continued to decline, even as surface temperature began to increase again in the remaining frozen and thawing seasons, as well as partway into the active season (B–C). The Year 1 minimum MA28 methane flux of 0.1 mg C m$^{-2}$ h$^{-1}$ occurred during the active season (point E, August 2015). During the active season, relationships between surface temperature and MA28 methane fluxes were highly variable, showing both positive and negative correlations. These data from Year 1 that show in parts inverse relationships between methane flux and surface soil temperature (relative to Year 2 and Zona et al. (2016)) suggest that, under certain conditions, surface soil temperature alone cannot always reliably predict seasonal methane flux patterns. In fact, that some of the highest methane

emission observed during the period of coldest surface soil temperature in the Year 1 cold season is in direct contrast to the strong temperature-dependence of microbial activity reported by Zona et al. (2016).

Figure 2c,d similarly display relationships of MA28 net ecosystem respiration with MA28 surface soil temperature, whereby ecosystem respiration is approximated as carbon dioxide fluxes during periods when incoming PAR is less than 5 $\mu$mol m$^{-2}$ s$^{-1}$.
Note that this approach only provides an upper boundary for heterotrophic microbial activity as autotrophic respiration by plants also contributes to observed carbon dioxide fluxes (Hicks Pries et al., 2015). These panels show that, in both years, MA28 respiration fluxes decreased rapidly during the onset of both freezing seasons as soil surface temperature cooled, from 0.3 to 0.0 g C m$^{-2}$ h$^{-1}$ (Year 1) and 0.2 to 0.0 g C m$^{-2}$ h$^{-1}$ (Year 2). As surface soils cooled further during the frozen seasons, respiration fluxes remained low, with Year 2 frozen season MA28 respiration fluxes decreasing from 0.04 to 0.01 g C m$^{-2}$ h$^{-1}$,
whilst in Year 1 these values were more variable, declining from 0.1 g C m$^{-2}$ h$^{-1}$ to near-zero before increasing again to 0.05 g C m$^{-2}$ h$^{-1}$. Respiration fluxes increased in both years prior to and during the thawing periods as surface soils warmed. It is noteworthy that the beginning of these increases coincided with turning points in the MA28 methane flux-surface temperature relationship (points B and G). In the active season, as discussed previously, Year 2 showed higher net carbon dioxide emission than Year 1. Similarly, since surface soil temperatures were much warmer in the Year 1 cold season, heterotrophic
respiration in this upper oxic soil region remained high relative to Year 2 leading to higher cold season cumulative carbon dioxide losses (Table 2). The strong relationships between cold season ecosystem respiration fluxes and surface soil temperature, and the relative similarity between the two years, is consistent with patterns reported in previous research (Lüers et al., 2014; Euskirchen et al., 2012; Björkman et al., 2010), and largely explain differences in the temporal trends of ecosystem respiration flux between the two years. This result suggests that changing heterotrophic microbial respiration in the upper soil region is
not a suitable explanation for the differences in methane flux–surface soil temperature relationships observed between Years 1 and 2.

### 3.4 Regression tree approach to seasonal methane flux dynamics

As discussed above, methane fluxes are largely dependent on microbial processes that compete in outcome (i.e. methanogenesis vs. methanotrophy), yet show similar environmental dependencies (Le Mer and Roger, 2001). Previous soil methane studies
have shown a range of controlling factors on microbial activity related to both methanotrophic and methanogenic activities, including temperature, water table depth, oxygen availability and Eh, soil organic matter content, soil pH, soil texture and soil mineralogy, though soil temperature and water table depth are often identified as the major of these controlling factors (Le Mer and Roger, 2001; Yvon-Durocher et al., 2014; Gulledge et al., 1997). We employed a regression tree approach (Sachs et al., 2008) to explore non-linear relationships between observed methane fluxes and variables identified in the literature
as influential to methanotrophic/methanogenic activity. Of the known variables, soil organic carbon content, pH, texture and mineralogy cannot explain changes in fluxes over short time periods and hence were not included. Additionally, pore-space oxygen concentrations were not included in the analysis since it was measured only in the upper 40 cm and remained oxic throughout the entire study period. For the regression tree, we hence used soil temperature data from the surface to 150 cm depth, as well as surface VWC. Daily values were chosen in order to reduce the influence of diel variability, with net daily

sums (in the case of methane fluxes) and mean daily values (for temperature and water content) as inputs into the model. The outcome of this analysis is shown in Fig. 3a, along with the time series of net daily methane fluxes used to build the model (Fig. 3b). Horizontal lines and coloured shading in Fig. 3b show mean ± one standard deviation of the input methane flux data, as grouped by the predictive model. The predictive capability of this model, tested against a randomly-selected validation set representing 20% of the available input data, shows an $R^2$ value of 0.69 ($p <$0.001).

The two variables that most effectively cluster methane flux values within the hyper-dimensional data space are soil temperatures measured at 100 cm and at 10 cm depths. Critically, these two temperature variables separate, and likely explain, substantial methane flux differences observed during the frozen seasons between Year 1 and Year 2. Specifically, the frozen season in Year 2 was largely represented by regression tree outcomes when temperatures at 100 cm soil depth were below -2.4 °C, marked in Fig. 3b in blue and red. This is the only season during the study period when 100 cm soil temperature fell

below this threshold (see also Fig. 4). Measurements throughout the active layer by Gebauer et al. (1996) at nearby Imnavait Creek suggest that it is highly likely that soils at these depths are anoxic and thus methanogenesis is the dominant relevant microbial process taking place here. In contrast to Year 2, the Year 1 frozen season was largely separated from the remainder of the study period (with 100 cm soil temperatures above -2.4 °C) when 10 cm temperatures were simultaneously below -0.6 °C (purple and green shading). During this season, methane emission values were amongst the highest observed throughout the

entire study period. As discussed previously, our own soil pore gas measurements show that these surface soils are oxic and thus methanotrophy is here likely the dominant relevant microbial process. It must be noted that these temperature thresholds do not represent mechanistic limits but instead the most effective clustering of observed data, based on the chosen environmental parameters. Even so, the reasonably good predictive capability of this model provides strong evidence that a critical reason for strong flux differences between frozen season methane fluxes was differences in deep soil temperature between the two

415 years. More precisely, these first two results from the regression tree analysis show a threshold temperature value at a depth of approximately 100 cm that is linked to low methane emission in Year 2, and an additional threshold temperature value at a depth of approximately 10 cm that is linked with high frozen season emission in Year 1.

    A decrease in methanogenesis below a temperature threshold around -2.4 °C as suggested by the model is in reasonable agreement with several experimental laboratory studies. Incubation studies investigating the temperature dependency

of methanogenesis in Arctic soils have shown that it can take place at sub-zero temperatures, though at greatly reduced rates. Rivkina et al. (2004) reported substantial methane production at temperatures of -1.8 °C in Siberian permafrost soils, as well as methane production in these soils at temperatures as low as -16.5 °C, though at a rate 100 times lower than at -1.8 °C. Similarly, Panikov and Dedysh (2000) observed minor methane emission ($\sim$0.1 mg C dm$^{-3}$ h$^{-1}$) from Siberian peat bog soils at -20 °C that increased by an order of magnitude after thawing. Chowdhury et al. (2015), using soils from Barrow, Alaska, showed

evidence of substantial methanogenesis in organic and mineral active layer soils kept at 4 and 8 °C, yet this was not observed in permafrost soils or in active layer soils kept at -2 °C. Similarly, for methanotrophic activity, temperature dependencies have also been observed, again with lower microbial activity reported at lower soil temperatures. Jørgensen et al. (2015) reported an exponential relationship between temperature and methane uptake in unsaturated Arctic tundra soils. At 18 °C they observed a deposition flux of 192 $\mu$g C m$^{-2}$ h$^{-1}$ – this decreased to 24 $\mu$g C m$^{-2}$ h$^{-1}$ in soils kept at -4 °C. Richter (2019) also observed

a temperature-related decrease in methane oxidation in A- and B-horizon soils sampled near Toolik Lake, to below detection
levels at temperatures below -2 °C. Based on this evidence and our regression tree analysis, we suggest that the separation of
methane fluxes in the Year 2 frozen season (the period with the lowest methane fluxes across the entire study period) is linked
to an inhibition of methane production due to low soil temperatures in deep, anoxic soil horizons. Further, we suggest that the
separation of methane fluxes in the Year 1 frozen season (the period with some of the highest methane fluxes across the entire
study period) is linked to an inhibition of methane oxidation due to low soil temperatures in oxic, surface soil horizons.

   The full time series of interpolated soil temperature profile measurements at Toolik Field Station is shown in Fig. 4, along
with the -2.4 °C isotherm identified in the first grouping of the regression tree analysis. These profile data show clearly that the
deeper soil horizons never reached the cold temperatures in the Year 1 frozen season that they did in the Year 2 frozen season –
in fact, the -2.4 °C isotherm did not move below 70 cm depth in Year 1. Taking as a starting point the only methane production
rate in the aforementioned laboratory studies given per soil volume (0.1 mg C dm$^{-3}$ h$^{-1}$ at -20 °C, Panikov and Dedysh,
2000), the 80 cm of soil below this depth known to be above -2.4 °C could presumably sustain a methane production rate on
the order of 80 mg C m$^{-2}$ h$^{-1}$. The MA28 methane flux (black line, lettering corresponds in time to that given in Fig. 2) shows
cold-season decreases that accompany cold temperature pulses into the deeper soil horizons. This is most readily seen in Year
2 (F–G) with the greater contrast in soil temperatures, though it is also noted that in Year 1 the onset of a decrease in methane
emission (B–C) corresponds in time to a lowering of the -2.4 °C isotherm and a still-perceptible cold temperature pulse to lower
soil horizons. This highlights that the limits identified in the regression tree analysis (i.e. -2.4 ° at 100 cm depth) are not claimed
to be mechanistic, yet they still provide valuable insight into competing methanogenic/methanotrophic processes within the soil
profile. Further, whilst soil methane concentrations were not measured deep enough to confirm methane production at 100 cm
depth, the sustained methane gradient in the upper 40 cm of the organic profile during the Year 2 cold season, and especially the
450 Year 1 cold season, provide supporting evidence for production of methane below this surface layer. Frozen season differences
in the MA28 respiration flux (green line) and, particularly, the ratio of methane flux to respiration (yellow line) further reiterate
the disconnect between methane production and respiration that was highlighted in the discussion surrounding Fig. 2.

   The shading in Fig. 3 shows that, whilst cold season methane fluxes could not be grouped together within the hyper-
dimensional data space for Years 1 and 2, freezing and thawing periods largely were. This suggests that the remaining seasonal
methane flux dynamics can be related to the balance of continued methanogenesis at depth and methanotrophic activity near the
surface. Specifically, predicted methane fluxes outside the frozen seasons (orange, yellow, and brown shading) are separated
according to temperatures at 150 cm and 20 cm, yet follow a similar pattern whereby colder temperatures at depth (suggesting
inhibited methanogenesis) and warmer surface temperatures (suggesting enhanced methanotrophy) jointly lead to smaller pre-
dicted methane fluxes (see also Fig. 4). One exception to these general patterns is during active seasons at times when surface
VWC is greater than 0.65. During these periods the largest mean methane fluxes (as well as the largest methane flux variability)
of any grouping were observed (pink shading). This observation is consistent with studies reporting increased methanogenic
activity relative to methanotrophy under water-saturated soil conditions due to reduced oxygen diffusivity and highly reducing
conditions in otherwise oxic surface soils (Le Mer and Roger, 2001). Evidence of this reduced oxygen diffusivity, as well as in-
hibition of gas diffusion through the soil profile, can be seen in the soil pore gas measurements in Fig. S3, where melting ice in

the Year 2 thawing season resulted in a sharp decrease in soil pore oxygen concentration, as well as a build-up of methane and carbon dioxide concentrations in the upper 40 cm. Flooding of the sample inlets unfortunately precluded the collection of any such evidence in the Year 1 thawing season. Decreased gas diffusivity during these periods likely contributed to a suppression of the methane flux, which were amongst the lowest seen throughout both years (leaf group 6 in Fig. 3).

### 3.5 Implications for Arctic methane fluxes

Considerable debate exists over the potential future of methane fluxes in the Arctic tundra under future climates (Sweeney et al., 2016). Hydrologic modelling under IPCC forecasts by Hay and McCabe (2010) predicted warmer air temperatures with greater precipitation, leading to the suggestion that methane fluxes may increase as labile carbon becomes available due to permafrost thaw. Warming experiments undertaken in the field in Alaska have also shown that warmer and wetter soils resulting from increased snow cover emit considerably more methane during the active period (Natali et al., 2015). Zona et al. (2016) stated

that cold season fluxes made up to 64% of their reported annual methane emission, due to relatively low but consistent emission over a large portion of the year. Our Year 2 cold season methane emission accounts for only 54% of the annual total, yet for this study site is still greater than the cold season total given by Zona et al. (2016) for their study sites (2.0 g C m$^{-2}$ here compared to 1.7 $\pm$ 0.2 g C m$^{-2}$). It is significant that Ivotuk, the upland tundra site in this study, exhibited the largest cold season, and the largest annual, net methane emissions. Euskirchen et al. (2017), in their four-year study of methane exchange

at Imnavait Creek (another upland tundra site), reported lower net cold season methane emission of 1.4 $\pm$ 0.3 g C m$^{-2}$, yet note that they only had partial zero curtain (freezing season) data for one year. The methane emission dynamics we observed in Year 2 suggest that this upland tundra bioclimate region may be significant for carbon cycling, particularly during the cold season. Our Year 1 data further show that cold season methane emission can account for an even greater percentage of the annual budget and that, under certain conditions, cold season emission fluxes are among the highest throughout the year – as

high as peak active season emission from saturated soils. Cold season methane fluxes are also subject to significant inter-annual variability. We further provided evidence that particularly high cold-season methane emission occurs when deep soil horizons are insulated and temperatures remain above the point where methanogenesis is efficient, while cold surface soil temperatures simultaneously minimise methanotrophic activity.

Modelled forecasting of Arctic methane fluxes is typically undertaken using air temperature data, due to its relative ease of

490 measurement and prediction, and the assumption that air temperature is closely linked to soil temperature (Riley et al., 2011; Koven et al., 2013; Zhu et al., 2014). An analysis of a 29-year record at Barrow, Alaska, however, showed no correlation between increasing air temperature and methane concentration anomaly (Sweeney et al., 2016), suggesting that air temperature is an inadequate variable for predicting methane fluxes. Air and soil temperature measurements at Toolik Field Station taken since 1988 (Fig. S7) show that, whilst Year 1 was the shortest winter (defined here once more as the period where MA28

air temperatures <0 °C) on record, both Year 1 and Year 2 were unusually warm and similarly ranked in regards to total FDDs (3rd and 4th warmest on record, respectively). Similarly, minimum cold season soil temperatures in the upper (20 cm) and deep (100 cm) horizons were unusually warm in both years, comparative to the long-term record (2nd and 3rd highest values, respectively). Given the relative similarity in the temperature anomaly of both years compared to the last 28, the large

differences in cold season methane emission (2 times larger in Year 1 compared to Year 2) is unlikely related to a simple linear
relationship with increasing air, or even soil, temperatures.

Instead, we suggest the presence of a deep soil temperature *threshold* in anoxic horizons above which cold season methano-
genesis – and hence net methane emission – remains high. Climatologically, there was little difference between Years 1 and
2 in terms of cold season FDDs, or minimum soil temperatures, relative to the previous 28 years. Yet, as suggested in the re-
gression tree analysis and the temperature profiles in Fig. 4, any such temperature threshold was not crossed in the Year 1 cold
season, allowing methanogenesis to continue relatively unabated. Snow profile measurements (Fig. S8) show that, in addition
to both winters experiencing relatively warm air temperatures, deeper snow in Year 1 likely insulated the underlying soil such
that anoxic soil horizons cooled at a much slower rate. If, as has been predicted (e.g. Hay and McCabe, 2010), the Arctic
continues to warm and precipitation increases, high methane emission winters will likely become more prevalent in the future,
particularly also if enhanced summertime warming pulses penetrate deeper in the soil profile. Our observations highlight the
need for more sophisticated modelling of temperature regimes in the forecasting of methane emission. More importantly, we
suggest that the increasing number of year-round ecosystem flux measurement sites operating in Arctic regions should monitor
soil temperatures throughout the entire active soil region, rather than limit observations to the upper surface horizons. This is
particularly important for large-scale soil monitoring networks such as the Soil Climate Analysis Network (SCAN), the outputs
from which are important for enabling gridded modelling products for quantifying regional-scale carbon fluxes. Temperature
data throughout the entire active soil profile, preferably in conjunction with estimates of soil redox conditions, would help to
further elucidate the competing microbial processes that drive methane fluxes at the surface.

## 4    Conclusions

Year-round measurements of ecosystem-scale methane and carbon dioxide fluxes were undertaken at Toolik Field Station in
the Alaskan Arctic over two years. Annual carbon dioxide exchange budgets suggest that these observations are representative
of wet sedge tundra, with seasonal patterns that are characteristic of the Alaskan North Slope generally. Net methane and
carbon dioxide fluxes in the Year 2 cold season (2.0 g C m$^{-2}$ and 294 g C m$^{-2}$ respectively, over 269 days) were similar in
magnitude to values reported in similar studies, and positive correlations between surface soil temperature and methane were
observed as previously reported by Zona et al. (2016). Year 1 cold season net methane and carbon dioxide fluxes, however,
were 100% and 38% higher, over a shorter cold season (22 days shorter). Relationships between respiration fluxes and surface
soil temperature were similar between years and with those reported in the literature, suggesting that warmer soil temperatures
in the oxic surface horizon can largely explain the differences in annual cold season carbon dioxide budgets between the two
years. Methane flux and surface soil temperature, by contrast, showed almost reversed relationships between the two years,
suggesting that surface soil temperature was not always sufficient to explain methane emission dynamics over the course of
this study.

Whilst cold season soil temperatures and FDDs were similar across both years (relative to the 28-year record), we observed
that deeper snow pack in Year 1 led to significantly warmer soil temperatures, particularly in the deeper portion of the active

soil profile. A regression tree analysis shows that high Year 2 frozen season methane fluxes were clustered from other data along the deep (100 cm) temperature axis at a threshold of -2.4 °C, suggesting inhibited methanogenesis in deeper, anoxic soil horizons. The highest cold season fluxes (among the highest of the two-year study) were observed during Year 1, when deep
soil temperatures remained above this threshold whilst surface temperatures were simultaneously below -0.6 °C, suggesting limited methanotrophy in upper soils being unable to offset methane production at depth. From our data we cannot reliably state that these thresholds represent mechanistic limits, only that they highlight a pattern of temperature dynamics between the upper, oxic layer and the deeper, anoxic layer that are key to controlling surface methane fluxes via their limiting influence on the competing processes of methanotrophy and methanogenesis. These temperature dynamic patterns further explain methane
flux dynamics outside of the frozen season during both years. Our results suggest that high cold season methane emission may be associated with warmer atmospheric temperatures and deeper snowpacks, and highlight a need for measurement and modelling of soil temperatures throughout all seasons, and throughout the entire active soil profile. Such expansion in observation capacities will allow more accurate prediction of potential changes in the annual methane exchange budget in Arctic tundra regions.

*Data availability.*  Data of this project, including atmospheric mercury, carbon dioxide and methane concentrations and mercury concentration measurements in vegetation, soils and snow are archived with NSF's Arctic Data Center (https://arcticdata.io/), DOI: 10.18739/A21Z41S5S.

*Author contributions.*  Dean Howard led the analysis of data and manuscript preparation. Yannick Agnan, Detlev Helmig, Yu Yang and Daniel Obrist designed the field measurement campaign. Yannick Agnan, Detlev Helmig and Daniel Obrist conducted field measurements. All authors contributed to writing of the manuscript.

*Competing interests.*  The authors declare that they have no competing interests, financial or otherwise.

*Acknowledgements.*  This study was funded by a by the U.S. Department of Energy (DE-SC0014275) and the U.S. National Science Foundation Office of Polar Programs (# OPR 1304305 and 1739567). The authors would like to acknowledge the help of Toolik Field Station for measurement support and the Toolik Field Station Environmental Data Center for providing soil temperature data. We also thank Donald A. (Skip) Walker for allowing the use of the Toolik field maps reproduced in the supplemental material. We wish to also thank the anonymous
reviewers whose constructive feedback contributed significantly to the development of the manuscript.

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

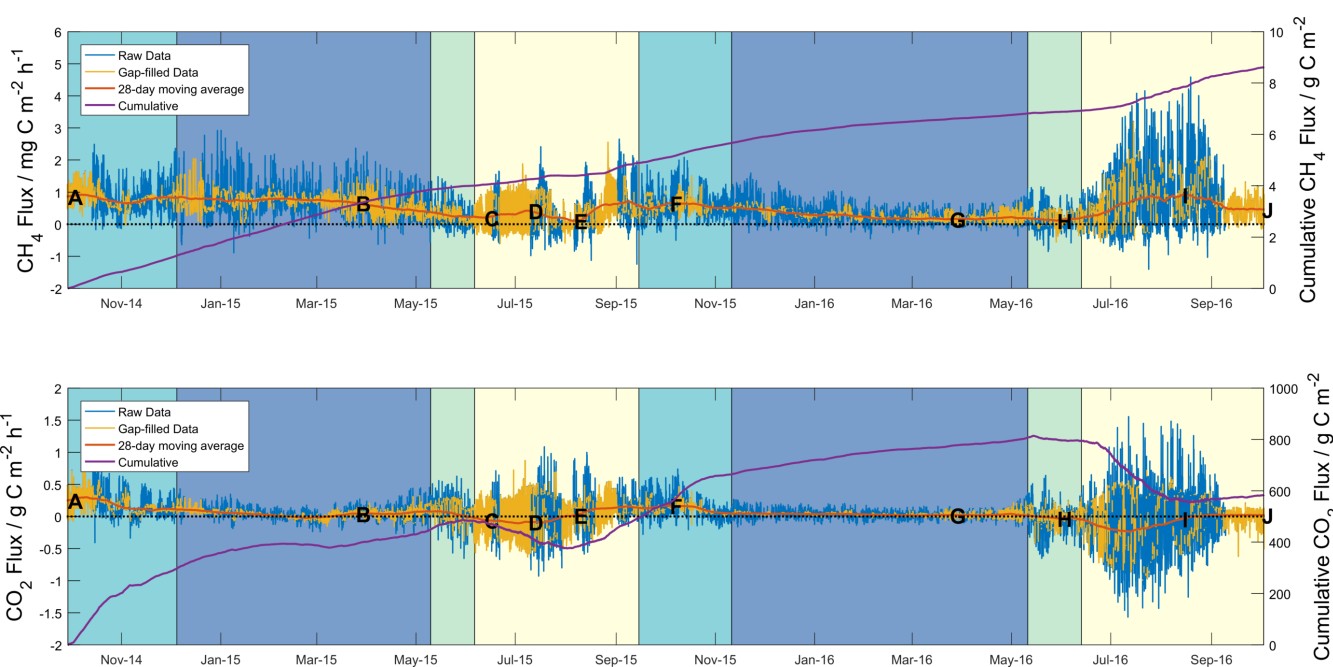

**Figure 1.** Half-hourly measured (blue lines) and gap-filled (yellow lines), and cumulative flux data (purple lines) for methane (upper panel) and carbon dioxide (lower panel). 28-day centred moving averages (red lines) have been included based on half-hourly flux data. Shading shows seasons (as defined in the text) for both years beginning with freezing (turquoise), then frozen (blue), thawing (green) and active (yellow). Lettering corresponds to that defined in the caption of Figure 2.

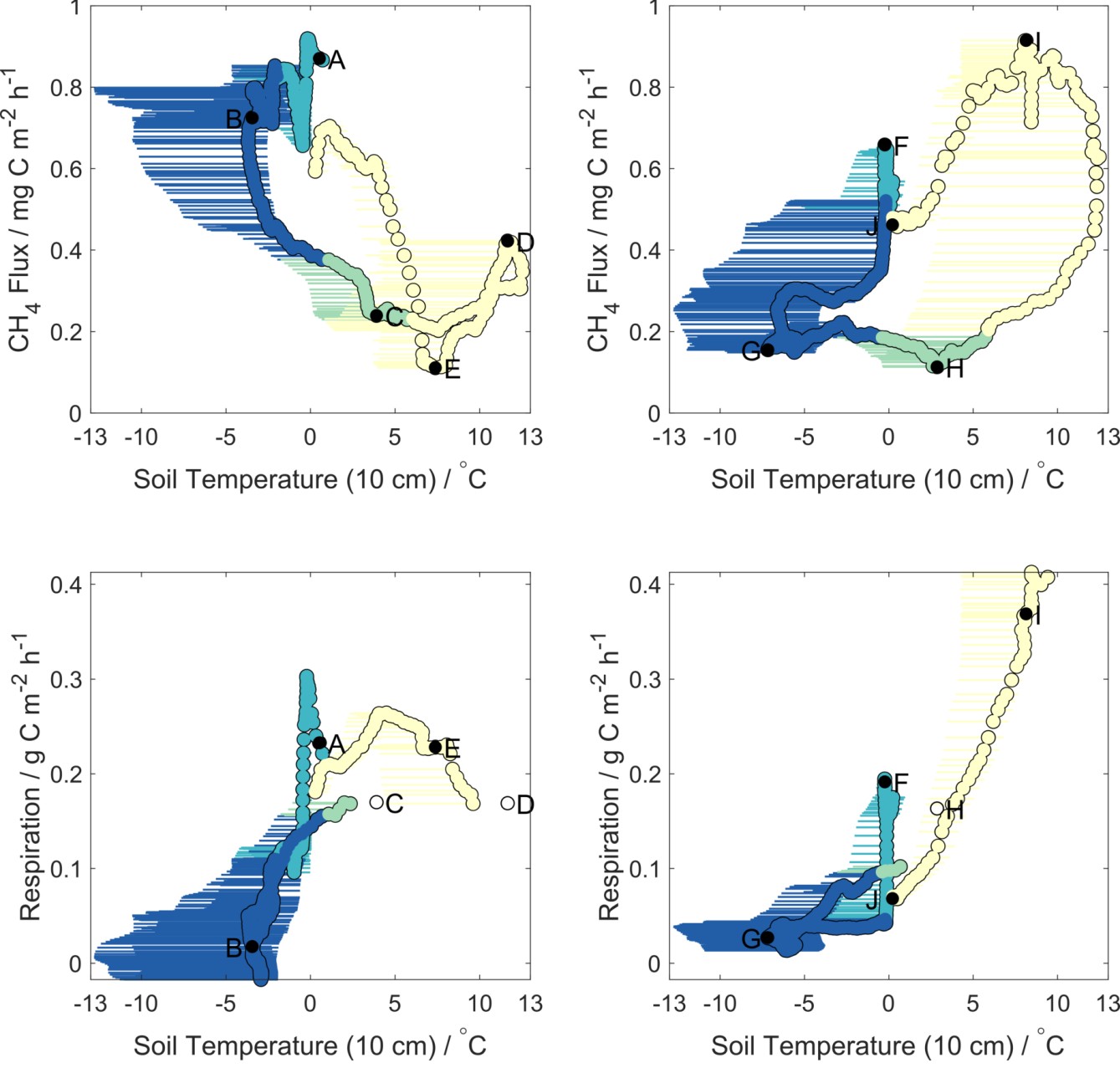

**Figure 2.** Methane (upper panels) and respiration (lower panels) fluxes against soil temperatures for Year 1 (left panels) and Year 2 (right panels). Horizontal error bars represent the range of soil temperatures across all four sampling pits, circles represent values used in decision tree analysis (average of both Toolik Field Station profiles). Colours correspond to seasons as in Fig. 1. Lettering on methane flux plots are sequential in time and correspond to those shown in Figs. 1 and 4. Lettering on respiration plots are given for the same times as those in methane plots. Open circles represent times for which respiration data are missing – respiration values here are linearly interpolated in time between the closest known values.

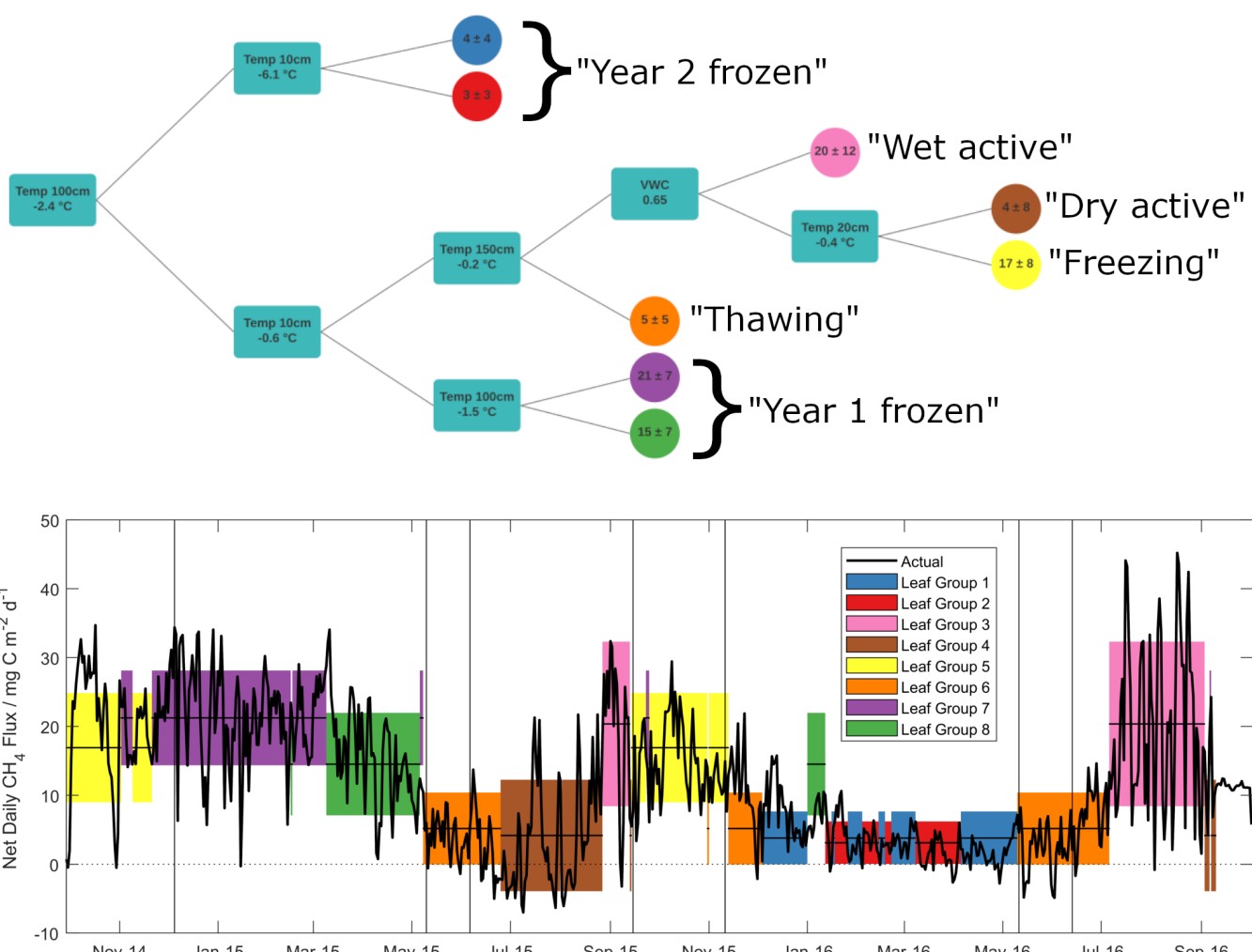

**Figure 3.** Outcome of the regression tree analysis (upper panel), giving decision steps and outcomes (mean ± standard deviation in mg C m$^{-2}$ d$^{-1}$) of selected methane emission data. Turquoise squares give the variable and thresholds around which decisions are made – lines pointing upwards correspond to values above this threshold and lines pointing down to values below. Lower panel shows net daily methane flux data (black line), along with means (horizontal lines) and standard deviations (shaded regions) of input data, as grouped by the regression tree in upper panel. Note that all methane flux data are net daily sums.

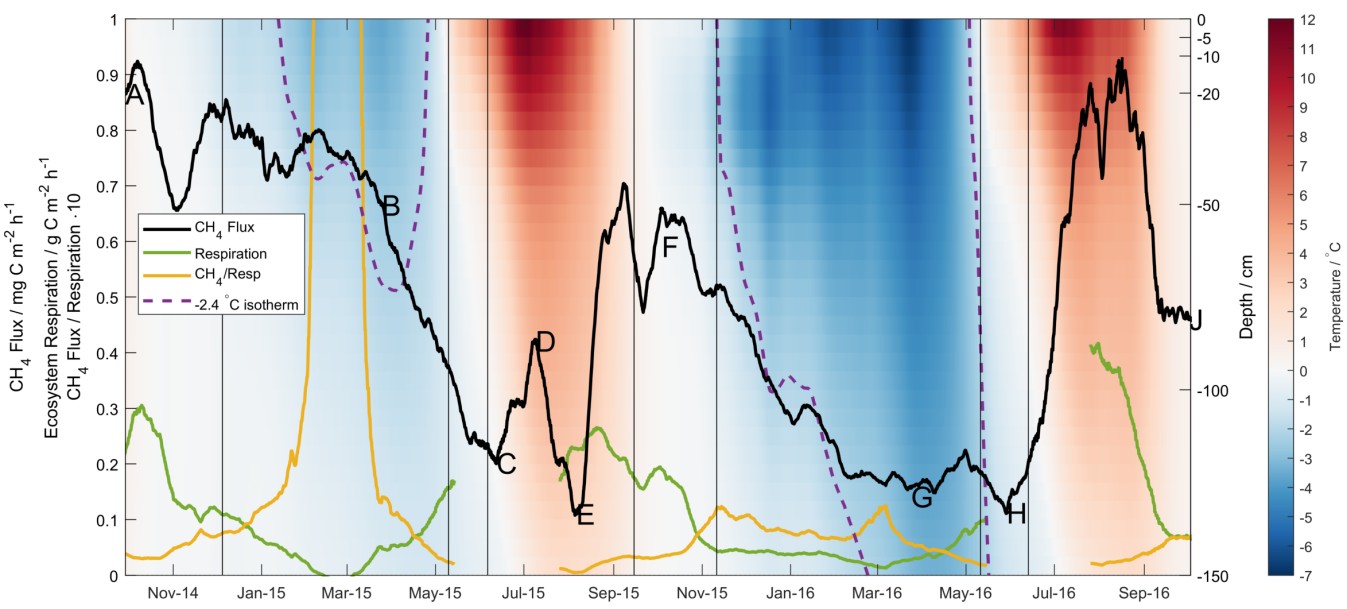

**Figure 4.** 28-day moving average methane (black line) and ecosystem respiration (green line) fluxes, along with the ratio of the two (yellow line). Lettering corresponds in time to that given in Fig. 2. Shading gives soil temperature as measured at Toolik Field Station according to depth (right axis). Purple dotted line shows the depth of the -2.4 °C isotherm.

**Table 1.** Overview of methane fluxes split according to season. Values in parentheses in cumulative columns are percentages of the total for that year. Daily differences are the differences in mean daily fluxes for that season between the two years, with $p$-statistics from Student's two-sample $t$-test giving the significance with which the null hypothesis (values come from distributions with same mean) can be rejected.

| | Year 1 29 Sep 2014 – 14 Sep 2015 | | | Year 2 15 Sep 2015 – 03 Oct 2016 | | | Daily Difference |
|---|---|---|---|---|---|---|---|
| | Duration days | Cumulative $\mathrm{g\,C\,m^{-2}}$ | Daily $\mathrm{mg\,C\,m^{-2}\,d^{-1}}$ | Duration days | Cumulative $\mathrm{g\,C\,m^{-2}}$ | Daily $\mathrm{mg\,C\,m^{-2}\,d^{-1}}$ | Value ($p$) $\mathrm{mg\,C\,m^{-2}\,d^{-1}}$ |
| Freezing | 66 | 1.3 (27%) | $19 \pm 4$ | 56 | 0.8 (21%) | $14 \pm 4$ | 5 (<0.001) |
| Frozen | 155 | 2.5 (51%) | $16 \pm 5$ | 181 | 1.1 (30%) | $6 \pm 3$ | 10 (<0.001) |
| Thawing | 26 | 0.2 (4%) | $6 \pm 2$ | 32 | 0.1 (3%) | $3 \pm 2$ | 3 (0.003) |
| Active | 100 | 0.9 (18%) | $9 \pm 8$ | 111 | 1.7 (46%) | $15 \pm 9$ | 6 (<0.001) |
| Total | 347 | 4.9 | | 380 | 3.7 | | |

**Table 2.** As for Table 1, for carbon dioxide fluxes. Seasonal percentages are represented as the absolute value of net seasonal exchange as a proportion of the annual total. As seasonal net totals are bi-directional, these percentages do not necessarily add to 100.

| | Year 1 29 Sep 2014 – 14 Sep 2015 | | | Year 2 15 Sep 2015 – 03 Oct 2016 | | | Daily Difference |
|---|---|---|---|---|---|---|---|
| | Duration days | Cumulative $\mathrm{g\,C\,m^{-2}}$ | Daily $\mathrm{g\,C\,m^{-2}\,d^{-1}}$ | Duration days | Cumulative $\mathrm{g\,C\,m^{-2}}$ | Daily $\mathrm{g\,C\,m^{-2}\,d^{-1}}$ | Value ($p$) $\mathrm{g\,C\,m^{-2}\,d^{-1}}$ |
| Freezing | 66 | 293 (60%) | $4 \pm 3$ | 56 | 167 (188%) | $3 \pm 2$ | 1.5 (0.001) |
| Frozen | 155 | 148 (31%) | $1 \pm 1$ | 181 | 142 (106%) | $0.8 \pm 0.5$ | 0.2 (0.04) |
| Thawing | 26 | 30 (6%) | $1 \pm 2$ | 32 | -15 (17%) | $0 \pm 1$ | 1.6 (<0.001) |
| Active | 100 | 14 (3%) | $0 \pm 3$ | 111 | -211 (237%) | $-2 \pm 3$ | 2.0 (<0.001) |
| Total | 347 | 485 | | 380 | 89 | | |