# Peer review of "Environmental controls on ecosystem-scale cold season methane and carbon dioxide fluxes in an Arctic tundra ecosystem."

_Biogeosciences, 2019_

## Referee Comment (RC1) · Anonymous Referee #1 · 14 Dec 2019

The manuscript by Howard et al. reports two years of ecosystem-scale methane fluxes in an Alaskan tundra ecosystem measured with the aerodynamic gradient approach. They find large contributions of wintertime CH4 emissions to annual emissions and pronounced differences in flux magnitudes between the two years. They propose that deep and near-surface soil temperature dynamics cause these differences in CH4 emissions with warm deep soil temperature boosting methanogenesis and cold near-surface temperatures inhibiting methanotroph activity. The manuscript aims to improve our current understanding of methane emissions in tundra regions. It addresses an important topic and contributes methane flux measurements for a region, where such measurements are still rare. The authors discuss an interesting hypothesis about the

origin of the observed interannual differences in methane emissions. However, I have two major comments regarding the manuscript that the authors should consider addressing. First, I am not sure if an ecosystem-scale flux measurement approach only is enough to support the authors' hypothesis. Flux measurements represent an important tool to derive greenhouse gas budgets and to characterise temporal greenhouse gas flux dynamics. However, soil profiles of methane concentrations would be needed to support the authors' hypothesis. Such measurements can provide information on where in the soil profile methane is produced and where it is consumed. High methane concentration in deeper soil layers would provide additional evidence for substantial methane production during winter conditions. Without this information, the authors' conclusions remain speculative. Second, the authors use the aerodynamic gradient approach to calculate methane fluxes between surface and atmosphere. How does this approach compare to the widely used eddy covariance method? What are the limitations of this approach? The manuscript mainly focusses on wintertime fluxes. Stable atmospheric conditions are characteristic for tundra ecosystems in the winter when snow cover is present. How would stability affect accuracy of the aerodynamic gradient approach? Equation 1 shows the stability-dependent similarity functions. How sensitive are calculated fluxes to these terms and is the effect the same for winter and summer? For eddy covariance measurements, a friction velocity threshold is usually applied to filter for period of low turbulence. Would such a threshold filter also apply to the aerodynamic gradient approach? The authors could consider quantifying these uncertainties and discussing potential implications on methane emission estimates.

Other comments

Line 13: Is there any evidence in the literature that sub-zero soil temperatures allow sufficient methane production to explain winter emission rates in this study?

Line 62: The authors mention here soil pore space methane and carbon dioxide concentrations, but these data are not presented in the manuscript.

Line 82: Should the unit rather be kg C m-2?

Line 91: Was only a zero calibration applied or also a span calibration?

Line 93-93: Could these concentration differences be used to derived uncertainties for the flux estimates? How much would a methane concentration bias of 0.001 $\mu$mol mol$-1$ affect methane emissions?

Line 120-121: Did the authors also account for potential effects of snow cover on displacement height?

Line 131: Which approach was used to gap-fill? Which function was used withing the R package?

Line 140: Here, and throughout the manuscript, comparisons could be supported by statistical test if possible (see for example t-test for snow depth).

Line 144: What is the response time of soil temperatures at 100 cm? How long does it take for a temperature pulse to propagate through the soil profile (see line 291)? Could it be that 100 cm winter soil temperature contains information from previous seasons?

Line 183: The soil respiration losses of about 0.5 kg C m-2 yr-1 seem very high to me (see also comparison with other tundra sites in the manuscript). Is there any particular reason why such high losses could be expected?

Line 185: The authors argue that methane production occurs deeper in the soil profile? Wouldn't it then be more intuitive to use deeper soil temperature time series to define transition seasons?

Fig. 2: Are methane emissions of 0.8 mg C m-2 h-1 reasonable for soil temperature below 0C? Methane production rates in the soil then must be at least of similar magnitude (i.e., in the absence of methane consumption in upper soil layers).

Line 324: The regression tree approach should be explained in the Methods section.

Line 340-341: Was the performance of the model equally good for winter and summer periods?

Line 360-377: Could these studies quantitatively support the temperature threshold at -2.4C?

Line 432: It is true that snow accumulation might increase in the tundra in a warming climate. However, melt periods during the cold winter period may become more frequent and lead to snow-free conditions during the winter. This could then lead to colder soil temperatures.

Line 437-439: The authors could discuss literature on wintertime methane concentration soil profiles if such studies exist.

---

## Referee Comment (RC2) · Anonymous Referee #2 · 15 Jan 2020

The paper "Environmental controls on ecosystem-scale cold season methane and carbon dioxide fluxes in an Arctic tundra ecosystem" by Howard et al. presents new year-round measurements and analysis of methane and carbon dioxide fluxes and environmental variables in an undersampled ecosystem type. Through well-reasoned and well-written description, the authors differentiate the impacts of soil temperature on microbial activity in the upper and lower portions of the active soil profile, specifically highlighting the role that unfrozen deep layer soil can have on the total methane emissions in Arctic tundra. This is an important insight, supported by in-situ data, that is worthy of rapid publication in Biogeosciences and may significantly impact future understanding of this system in a changing climate.

[Figure]

Specific minor comments and suggestions follow below:

1. The laboratory study in lines 43-46 seems a bit old to be the only one mentioned. Have there now been any more recent studies of these relationships? Perhaps the incubation studies on page 11 could be integrated into this introduction?

2. The additional measurements are clearly useful to have. More emphasis could be added at the end of the introduction relating to what sets this study location apart from those in Zona et al. 2016.

3. Is the gap-filling in line 133 applied with daily value for days with at least some PAR < 5? This is a bit unclear.

4. The large range cited for the wet sedge tundra site in line 227 is a result of a changing state at this location, rather than the representative variability of wet sedge itself.

5. Additional discussion could be added after line 400 relating to what happens to the methane flux in the case that high VWC soil freezes. Does frozen water present in the soil inhibit the gas transfer upward from the methanogens?

6. Perhaps toward the end of section 3.5 point out the importance of additional soil temperature information to improving gridded products, which are needed to fully quantify regional to pan-Arctic scale carbon fluxes.

7. Could the letter labels from Figure 2 be added to their appropriate time positions in Figure 1? This would better link the data during the description sections.

---

## Author Comment (AC1) · 1 May 2020

The authors wish to thank the anonymous reviewer for their time and for their constructive comments regarding the manuscript. We believe this feedback has had a large, positive impact on the outcome of the current manuscript. We present below the reviewer's comments, along with our responses and any changes made to the manuscript or supporting information in bullets. Line numbers correspond to the new version of the manuscript submitted along with these responses. Any changes made to the manuscript/SI are marked in blue within the respective document.

Reviewer 1: The manuscript by Howard et al. reports two years of ecosystem-scale

methane fluxes in an Alaskan tundra ecosystem measured with the aerodynamic gradient approach. They find large contributions of wintertime CH4 emissions to annual emissions and pronounced differences in flux magnitudes between the two years. They propose that deep and near-surface soil temperature dynamics cause these differences in CH4 emissions with warm deep soil temperature boosting methanogenesis and cold near-surface temperatures inhibiting methanotroph activity. The manuscript aims to improve our current understanding of methane emissions in tundra regions. It addresses an important topic and contributes methane flux measurements for a region, where such measurements are still rare. The authors discuss an interesting hypothesis about the origin of the observed interannual differences in methane emissions. However, I have two major comments regarding the manuscript that the authors should consider addressing.

First, I am not sure if an ecosystem-scale flux measurement approach only is enough to support the authors' hypothesis. Flux measurements represent an important tool to derive greenhouse gas budgets and to characterise temporal greenhouse gas flux dynamics. However, soil profiles of methane concentrations would be needed to support the authors' hypothesis. Such measurements can provide information on where in the soil profile methane is produced and where it is consumed. High methane concentration in deeper soil layers would provide additional evidence for substantial methane production during winter conditions. Without this information, the authors' conclusions remain speculative.

- This is a good point and one that we have stressed throughout the paper (see, for example, line 531). Our experimental design was developed to include measurements of soil methane concentrations, and in response to this and other comments from both reviewers, we have expanded our discussion to include these measurements. Though we only measured to 40 cm below the surface, these soil profile measurements provide supporting evidence of diffusion of methane from below the surface layer during the cold season. These measurements, and

the regression tree analysis, both support our hypothesis regarding cold season methane production at depth. Further, our observations remain novel, providing some of the first soil pore space and surface flux data for methane during the cold season in Alaska, broadening the range of Arctic tundra bioclimates studied during the cold season, and showing as-yet unpublished interannual variability in cold season methane fluxes (100% increase in methane emission between subsequent years).

- Included data on soil pore space methane data (Fig. S3) and discussion regarding this (see responses to below comments for details).

Second, the authors use the aerodynamic gradient approach to calculate methane fluxes between surface and atmosphere. How does this approach compare to the widely used eddy covariance method? What are the limitations of this approach?

- Included on line 87: "The aerodynamic gradient method was chosen for compatibility with a concurrent atmospheric mercury flux study (Obrist et al., 2017). This method has been shown to have greater variability compared to the more widely-used eddy covariance method over diel time scales (Muller et al., 2009) though with reasonable agreement over longer time scales, with the caveat that the concentration gradients are precisely quantified using high-precision gas analysers (Zhao et al., 2019; Fritsche et al., 2008; Karlsson, 2017)."

The manuscript mainly focusses on wintertime fluxes. Stable atmospheric conditions are characteristic for tundra ecosystems in the winter when snow cover is present. How would stability affect accuracy of the aerodynamic gradient approach? Equation 1 shows the stability-dependent similarity functions. How sensitive are calculated fluxes to these terms and is the effect the same for winter and summer?

- The Monin-Obukhov stability theory limitations did not exclude a lot of data. We have added additional information regarding this. See also response to comment

below regarding the friction velocity filtering – this had a larger impact, as well as a slight seasonal impact.

- Included in line 132: "that are well-represented for scalars within a range of Obukhov lengths between -2.5 and 2. Values outside of this range (highly stable or highly unstable) were filtered from analysis, resulting in a loss of 1.1% of available data."

- See also response to comment below.

For eddy covariance measurements, a friction velocity threshold is usually applied to filter for period of low turbulence. Would such a threshold filter also apply to the aerodynamic gradient approach? The authors could consider quantifying these uncertainties and discussing potential implications on methane emission estimates.

- Correct, a friction velocity threshold was applied, though this was not originally stated in the manuscript.

- Included in line 134: "Additionally, periods for which the friction velocity was below 0.08 m s$^{-1}$ (Muller et al., 2009) were excluded from analysis. This resulted in a loss of 7.1% of all available data, with a slight seasonal bias (7.1% of winter (DJF) data compared to 2.9% of summer (JJA) data)."

Other comments Line 13: Is there any evidence in the literature that sub-zero soil temperatures allow sufficient methane production to explain winter emission rates in this study?

- There is and these are discussed in the paragraph beginning line 413.

- No changes made.

Line 62: The authors mention here soil pore space methane and carbon dioxide concentrations, but these data are not presented in the manuscript.

- Included in line 95: "In addition to atmospheric trace gas sampling, a soil trace gas system consisting of four to six soil inlets in two vertical profiles allowed for determination of near-surface trace gas concentration gradients in soil pore air (Fig. S2). Data from these two soil inlet profiles were collected at two depths in Year 1 (10 and 40 cm), and three depths in Year 2 (10, 20 and 40 cm)."

Line 82: Should the unit rather be kg C m-2?

- Yes thank you, well noticed. The 'k' was accidentally dropped.

- Line 86: changed "density" to "stores" and inserted "k".

Line 91: Was only a zero calibration applied or also a span calibration?

- Span calibrations were also performed, using the internal calibration routine within the instrument. This information was missing in the original manuscript.

- Inserted into line 106: "Span calibrations were achieved using the internal calibration routine, as recommended by the manufacturer."

Line 93-93: Could these concentration differences be used to derived uncertainties for the flux estimates? How much would a methane concentration bias of $0.001 \mu molmol-1$ affect methane emissions?

- The concentration differences were very small compared to the average measured gradient and we have included information to show this.

- Included in line 109: "(or 0.025% of the mean concentration gradient)" and "(mean methane gradients were -4 $\pm$ 7 $\mu$mol mol$^{-1}$)" and "(0.1% of the mean concentration gradient)" and "(mean carbon dioxide gradients were 0.1 $\pm$ 0.2 mmol mol$^{-1}$)".

Line 120-121: Did the authors also account for potential effects of snow cover on displacement height?

- We did and we have included additional information to show this.

- Included in line 142: "Displacement height d was set as $0.7hc$ (canopy height, $\sim$0.2 m) during snow-free periods and at 0 m during snow-covered periods (Oke, 1983)."

Line 131: Which approach was used to gap-fill? Which function was used withing the R package?

- We have add additional information regarding the gap-filling method.

- Included in line 153: "MDSGapFill function (Reichstein et al., 2005) within the"

Line 140: Here, and throughout the manuscript, comparisons could be supported by statistical test if possible (see for example t-test for snow depth).

- We have expanded the number of t-tests for distributions of climatological data presented in the manuscript.

- Included in line 167: "Students t-tests could not reject the null hypothesis that mean values for these months in these years came from a different distribution than the climatological record ($p = 0.26$ and 0.68, respectively)."

- Included in line 171: "(Student's t-test, $p < 0.001$)"

- Included in line 172: "(Student's t-test, $p < 0.001$)"

Line 144: What is the response time of soil temperatures at 100 cm? How long does it take for a temperature pulse to propagate through the soil profile (see line 291)? Could it be that 100 cm winter soil temperature contains information from previous seasons?

- Good question and we believe that the temperature can propagate to these depths within a single season. This was already explored to some extent in Fig. 4, where we show interpolated temperatures for the study period. However, there was no information regarding the temperatures in the preceding summer (of 2014). This we have attempted to rectify by adjusting Fig. S5 to show air, 20 cm and 100 cm temperatures for the 2013–14 year. This new figure shows that all three temperatures were similar for 2013–14 than for both 2014–15 and 2015–16 (and for the whole measured climatology). Therefore, we believe that inter-seasonal effects are less important than intra-seasonal (i.e. the temperature pulses are significant on a $< 3$ month time scale).

- Included new Fig. S5 (now Fig. S6).

Line 183: The soil respiration losses of about 0.5 kg C m-2 yr-1 seem very high to me (see also comparison with other tundra sites in the manuscript). Is there any particular reason why such high losses could be expected?

- This number does at first instance appear quite high but we believe it to be valid. The reasons for this are discussed further in the paragraph beginning line 278 (net active season emission plus relatively high freezing season emission). We have added some extra discussion to justify our belief in this number.

- Included in line 282: "For Year 1 however, the active season sink was relatively short-lived, resulting in a net active season emission (0.9 g C m$^{-2}$ h$^{-1}$) that contributed to the large net annual emission for Year 1."

- Included in line 292: "In a latitudinal comparison from three Alaskan tundra sites, Grogan and Chapin (1999) reported significantly higher wintertime carbon dioxide efflux from Toolik, relative to Fairbanks (south of Toolik) and Sagwon (north of Toolik). This wintertime carbon dioxide efflux was correlated with warmer surface soil temperatures (quantified as 5 cm soil temperatures greater than -5 °C), which at Toolik were relatively high due to thermal insulation by a substantial early snowfall. The extended period of increased carbon dioxide emission in the Year 1 freezing season is likely also associated with the insulating effects of an early substantial snowfall and the associated warmer surface soil temperatures (explored in greater detail in the following section)."

Line 185: The authors argue that methane production occurs deeper in the soil profile? Wouldn't it then be more intuitive to use deeper soil temperature time series to define transition seasons?

- Unfortunately, no. This was attempted and did not yield direct causative links between the deep soil temperature and observed fluxes. We are here proposing that there is a potential synergistic relationship between the entire soil temperature profile (deep and upper) with the observed methane flux (see paragraph beginning line 395, and esp. line 493 and following paragraph). The definition of seasons according to upper soil temperatures presented in line 211 and later discarded was introduced in order to show that this definition (used previously in the literature) does not hold for this particular data set, thus building the case for investigation using the regression tree approach.

- No changes made.

Fig. 2: Are methane emissions of 0.8 mg C m-2 h-1 reasonable for soil temperature be-low 0C? Methane production rates in the soil then must be at least of similar magnitude (i.e., in the absence of methane consumption in upper soil layers).

- We do believe that they are reasonable. For example, Zona et al. (2016) reported methane eddy covariance fluxes at one site (Ivotuk) that ranged from ∼1.5 g C m$^{-2}$ h$^{-1}$ to ∼0.5 g C m$^{-2}$ h$^{-1}$ for surface soil temperatures between 0.75 and -0.75 °C, then stayed at ∼0.5 g C m$^{-2}$ h$^{-1}$ for surface soil temperatures down to ∼-7 °C. Based on this, we don't think 0.8 g C m$^{-2}$ h$^{-1}$ for surface soil temperatures down to ∼-4 °C is unreasonable. The reviewer's point regarding methane production rates is a very good one and we have therefore included a simple calculation based on this thought experiment and some of the already-quoted literature.

- Included in line 418: (∼0.1 mg C dm-3 h-1)

- Included in line 434: "Taking as a starting point the only methane production rate in the aforementioned laboratory studies given per soil volume (0.1 mg C dm$^{-3}$ h$^{-1}$ at -20 °C, Panikov and Dedysh, 2000), the 80 cm of soil below this depth known to be above -2.4 °C could presumably sustain a methane production rate on the order of 80 mg C m$^{-2}$ h$^{-1}$."

Line 324: The regression tree approach should be explained in the Methods section.

- It should have but was not. We have included information regarding this approach in the methods section.

- Included in line 158: "Regression tree analysis (Sachs et al., 2008; De'ath and Fabricus, 2000; Breiman et al., 1984) was undertaken on 80% of observed flux data (20% test fraction) using the TreeBagger function in Matlab 2016a (Math-Works, Natick, MA, USA). 500 cross validations were ran with a minimum leaf

size of 1% of the training set size, with the tree with lowest mean squared error chosen as our predictive model."

Line 340-341: Was the performance of the model equally good for winter and summer periods?

- We used here all data from the two-year study period in our predictive model, as to do otherwise would introduce user bias, as well as bring to the fore once more questions about definitions of seasons. As such, in terms of the predictive capability that we quote in line 394, it would only be appropriate to treat the data set as a whole. We do, however, discuss deeply into the seasonal implications of the predictive model in paragraphs beginning line 395 and line 498, accompanied by Figure 3b.

- No changes made.

Line 360-377: Could these studies quantitatively support the temperature threshold at-2.4C?

- None of the mentioned studies gave a threshold of -2.4 °C or provided any sort of linear relationship between temperature and methane production that would allow us to calculate, or quantitatively verify, that -2.4 °C is a mechanistic threshold. We stress (e.g. lines 407, 442, 532) that our data set cannot quantifiably point to an exact temperature threshold but, rather, that the regression tree analysis, along with the mentioned laboratory studies, provide strong evidence to support the hypothesis that we put forward in this manuscript.

- No changes made.

Line 432: It is true that snow accumulation might increase in the tundra in a warming climate. However, melt periods during the cold winter period may become more frequent and lead to snow-free conditions during the winter. This could then lead to colder soil temperatures.

- This is also true but beyond the scope of our analysis. We were careful to state that any future Arctic scenario mentioned may take place, based on modelling efforts by others, and that the scenarios mentioned may have an impact on methane emission in line with our hypothesis (i.e. increasing emission if deeper soils stay warmer). Beyond this, we are not attempting to validate possible future Arctic scenarios.

- No changes made.

Line 437-439: The authors could discuss literature on wintertime methane concentration soil profiles if such studies exist.

- We are not aware of other wintertime methane concentration soil profile studies. In response to this and previous comments, we have included additional information and discussion regarding the methane soil concentration profiles taken in this study.

- No changes made beyond those mentioned in other responses.
* * *
[Figure]

**Fig. 1.**

---

## Author Comment (AC2) · 1 May 2020

The authors wish to thank the anonymous reviewer for their time and for their constructive comments regarding the manuscript. We believe this feedback has had a large, positive impact on the outcome of the current manuscript. We present below the reviewer's comments, along with our responses and any changes made to the manuscript or supporting information in bullets. Line numbers correspond to the new version of the manuscript submitted along with these responses. Any changes made to the manuscript/SI are marked in blue within the respective document.

Reviewer 2: The paper "Environmental controls on ecosystem-scale cold season

methane and carbon dioxide fluxes in an Arctic tundra ecosystem" by Howard et al. presents new year-round measurements and analysis of methane and carbon dioxide fluxes and environmental variables in an under sampled ecosystem type. Through well-reasoned and well-written description, the authors differentiate the impacts of soil temperature on microbial activity in the upper and lower portions of the active soil profile, specifically highlighting the role that unfrozen deep layer soil can have on the total methane emissions in Arctic tundra. This is an important insight, supported by in-situ data, that is worthy of rapid publication in Biogeosciences and may significantly impact future understanding of this system in a changing climate.

Specific minor comments and suggestions follow below:

1. The laboratory study in lines 43-46 seems a bit old to be the only one mentioned. Have there now been any more recent studies of these relationships? Perhaps the incubation studies on page 11 could be integrated into this introduction?

- The newest reference we could find that considered both rates of methanotrophy and methanogenesis (Schipper et al., 2014; doi: 10.1111/gcb.12596) is a development of thermodynamic theory that was applied to the Dunfield et al. (1993) data. Unfortunately, the references on page 11 refer to only one process or the other.

- No changes made.

2. The additional measurements are clearly useful to have. More emphasis could be added at the end of the introduction relating to what sets this study location apart from those in Zona et al. 2016.

- As suggested, we added some additional discussion here relating to expansion of year-round data sets in different study locations.

- Included in line 56: "(four on coastal plains and tundra dominated by sedges, grasses, and mosses within the northern coastal region surrounding Utqiaġvik and one over tussock-sedge dwarf-shrub, moss tundra at Ivotuk on the North Slope of the Brooks Range, Walker et al., 2005),"

- Included in line 70: "expand our coverage of year-round methane and carbon dioxide exchange data sets across different bioclimates and landscapes, as well as"

3. Is the gap-filling in line 133 applied with daily value for days with at least some PAR< 5? This is a bit unclear.

- We agree that this was a little unclear, thank you. The gap filling was actually applied on the half-hourly data and we have included information to explain this in the text.

- Included in line 155: "half-hourly"

4. The large range cited for the wet sedge tundra site in line 227 is a result of a changing state at this location, rather than the representative variability of wet sedge itself.

- We have added some additional discussion to reflect this point.

- Included in line 265: "The low end of the range of values quoted for wet sedge tundra (2 g C m$^{-2}$ h$^{-1}$, Euskirchen, et al., 2012) is based on a period when active season deposition largely balanced cold season emission; Euskirchen et al. (2017) report in their longer-term study of this wet sedge site a trend towards larger annual net emission values that are largely attributed to increasing cold season emission, with little trend seen for active season deposition. They note an

increase in September–December carbon dioxide emission of 1.34 g C m$^{-2}$ for each additional day of zero curtain (freezing season) length. Here, the observed difference was much larger, with an additional 126 g C m$^{-2}$ loss observed in Year 1 over a 10 day longer freezing season (Table 2)."

5. Additional discussion could be added after line 400 relating to what happens to the methane flux in the case that high VWC soil freezes. Does frozen water present in the soil inhibit the gas transfer upward from the methanogens?

- This is a good point and related to another that Reviewer 1 had regarding soil pore gas concentrations. In addition to the inclusion of soil pore gas measurements and the surrounding discussion, we have expanded the discussion at this point to refer specifically to diffusion inhibition.

- Included in line 458: "Evidence of this reduced oxygen diffusivity, as well as inhibition of gas diffusion through the soil profile, can be seen in the soil pore gas measurements in Fig. S3, where melting ice in the Year 2 thawing season resulted in a sharp decrease in soil pore oxygen concentration, as well as a build-up of methane and carbon dioxide concentrations in the upper 40 cm. Flooding of the sample inlets unfortunately precluded the collection of any such evidence in the Year 1 thawing season. Decreased gas diffusivity during these periods likely contributed to a suppression of the methane flux, which were amongst the lowest seen throughout the year (leaf group 6 in Fig. 3)."

6. Perhaps toward the end of section 3.5 point out the importance of additional soil temperature information to improving gridded products, which are needed to fully quantify regional to pan-Arctic scale carbon fluxes.

- We are not overly familiar with such gridded products, but we tried. . .

- Included in line 507: "This is particularly important for large-scale soil monitoring networks such as the Soil Climate Analysis Network (SCAN), the outputs from which are important for enabling gridded modelling products for quantifying regional-scale carbon fluxes."

7. Could the letter labels from Figure 2 be added to their appropriate time positions in Figure 1? This would better link the data during the description sections.

- They sure can!

- Included letters in Figure 1 and altered caption.

---

## Author Response (AR2)

The authors wish to thank the anonymous reviewer again for their time and valuable input to the development of the manuscript. We have included the reviewer's comments below in black, with our responses following in blue.

The authors state that it is important that "the concentration gradients are precisely quantified using high-precision gas analysers" to achieve high accuracy flux measurements. Would it be possible to use the information on system performance (lines 107-110) to calculate uncertainties for annual budgets (for example for line 204)? What if there would have been a small bias in concentration gradients between year 1 and 2? Adding an uncertainty range to annual CH4 emission estimates could provide further support that there was indeed a significant difference in CH4 emissions between years.

This is a good point, and we have quantified the uncertainty from our measured null gradients and included these in the manuscript.

Included in Line 111: One-way ANOVA tests performed on the line intercomparison data showed that methane null gradients were not significantly different throughout both years ($p = 0.03$), however this was not the case for carbon dioxide null gradients, with those from the final intercomparison (in December 2015) being significantly lower than the rest. Estimation of the cumulative uncertainty calculated from null gradient data (achieved by substituting ($C_2 - C_1$) in Eq. 1 with ($C_2 - C_1 + \varepsilon$), where $\varepsilon$ is the measured null gradient value), gave values of 0.5% and 0.2% for methane in Years 1 and 2, respectively. For carbon dioxide, the respective uncertainties were 8% and 47%.

Would it be possible to use the CH4 concentrations gradients (see Fig. 3) to estimate a ballpark figure for methane emissions (e.g. using some literature-based soil-gas diffusivities)? I think one of the unique aspects of this manuscript is that it pairs ecosystem-scale flux with pore gas concentration measurements. The authors extended the discussion of the pore gas concentration data, but could, in my opinion, use these data to further strengthen the manuscript.

This is certainly a valid desire, and one that we discussed and strove for at length during the initial stages of developing the manuscript. Unfortunately, arriving at suitable literature-based soil-gas diffusivities was prohibitively difficult due to the changing conditions of the substrate. For example, the soil became saturated with water at times (most notably thawing seasons), which could lead to an order-of-magnitude change in the soil-gas diffusivity (e.g. Hu et al., 2018; doi: 10.3390/app8112097). Further, during early freezing and thawing seasons, we noted large increases in soil pore-gas concentrations that we can only hypothesise were due to changes in soil-gas diffusivities related to freezing within the soil. Lastly, the presence of snow above the soil for most of the year, has a very large and uncertain impact on gas diffusivity, depending on crystal type, porosity, lensing, and wind speeds (e.g. Whelsky, 2017; https://search.proquest.com/docview/1989143200).

For these reasons, we decided that a simple model of gas diffusion would not be suitable for providing satisfactory insights, and unfortunately we do not have the capabilities within the group to create a more complex model. We would be thrilled to work with a researcher or group of researchers who do have such capabilities, and will be happy to share our data should anyone be interested.

No changes made.